# Properties of Putative APSES Transcription Factor AfpA in *Aspergillus fumigatus*

**DOI:** 10.3390/jof11090678

**Published:** 2025-09-16

**Authors:** Young-Ho Choi, Min-Woo Lee, Kwang-Soo Shin

**Affiliations:** 1Department of Microbiology, Graduate School, Daejeon University, Daejeon 34520, Republic of Korea; youngho1107@gmail.com; 2Soonchunhyang Institute of Medi-Bio Science, Soonchunhyang University, Cheonan 31151, Chungcheongnam-do, Republic of Korea

**Keywords:** *Aspergillus fumigatus*, APSES transcription factor, AfpA, asexual development, stress responses, virulence

## Abstract

*Aspergillus fumigatus* is a major opportunistic pathogenic fungus that causes invasive aspergillosis with high mortality rates in immunocompromised patients. APSES family transcription factors regulate fungal development and virulence, but the function of the putative APSES-type transcription factor AfpA (AFUA_5g11390) remains uncharacterized. To investigate the roles of AfpA in *A*. *fumigatus*, we constructed the Δ*afpA* mutant and performed phenotypic analyses, RT-qPCR analyses, and virulence studies. The Δ*afpA* mutant exhibited reduced vegetative growth but increased conidiation, with upregulation of asexual developmental regulators *brlA*, *abaA*, and *wetA*. AfpA positively regulated cAMP-PKA signaling, resulting in delayed conidia germination. Furthermore, the mutant responded differently to external stresses and displayed enhanced virulence in neutropenic mice. In conclusion, AfpA functions as a multifaceted regulator balancing growth, development, and pathogenicity in *A. fumigatus*.

## 1. Introduction

*Aspergillus fumigatus* is a major opportunistic fungal pathogen responsible for invasive pulmonary aspergillosis in immunocompromised individuals. This ubiquitous saprophytic mold is the most common cause of aspergillosis, with mortality rates ranging from 30% to 95% depending on the patient population [1]. The ability of *A. fumigatus* to cause disease is attributed to multiple virulence factors, including thermotolerance, rapid growth, small conidial size, and the production of various secondary metabolites [2]. Understanding the molecular mechanisms that regulate these virulence attributes is crucial for developing novel therapeutic strategies against this formidable pathogen.

APSES (Asm1p, Phd1p, Sok2p, Efg1p, and StuA) family transcription factors (TFs) play crucial roles in fungal growth, development, and virulence. StuA governs the upregulation of a discrete transcriptional program during the acquisition of developmental competence and is essential for proper conidiophore development [3]. RgdA functions as a master regulator that governs growth, asexual development, gliotoxin biosynthesis, and pathogenicity [4]. Other APSES members, such as MbsA, have been characterized and shown to influence fungal development and stress responses [5], while regulatory proteins like RgsA attenuate PKA signaling and virulence [6]. AfpA (Afp1 orthologous protein) contains a basic helix-loop-helix (bHLH) DNA binding motif, APSES domain. The APSES proteins are divided into four groups based on the structures of the APSES domain [7]. AfpA belongs in clade D, a clade that lacks functional definitions and is named as APSES-family protein 1. The coordinated action of these TFs underscores the complex regulatory networks controlling morphogenesis and pathogenicity in *A. fumigatus* [8].

Sequence analysis reveals a conserved KilA-N domain in AfpA, suggesting that AfpA may have a potential regulatory role. The KilA-N domain represents an ancient DNA-binding motif that originated in DNA viruses and was subsequently transferred to fungal genomes through lateral gene transfer events [9,10]. This domain has been implicated in transcriptional regulation and represents an evolutionary innovation in fungal cell biology [10]. The KilA-N family of TFs includes several subfamilies such as Swi4/6, Mbp1, Res1/2, Cdc10 (SMRC), Xbp1, APSES, and Bqt4, all of which share the conserved ancestral KilA-N DNA-binding domain [10]. This domain is homologous to the APSES DNA-binding domain found in fungi. Although KilA-N family members perform a wide range of biological functions, the APSES subfamily is the most extensively characterized and is recognized for its central role in regulating fungal development, secondary metabolism, and pathogenicity [7,10]. The presence of the KilA-N domain in AfpA, combined with its classification as an APSES family member, indicates that it may function as a transcriptional regulator with potentially important roles in *A. fumigatus* biology. However, the function of AfpA (Afu5g11390), a putative APSES-type TF, has not yet been studied in *A*. *fumigatus*.

Here, we characterize the biological role of AfpA in *A. fumigatus* by examining its effects on fungal development, stress response, and virulence. Our results provide novel insights into the functional diversity of APSES TFs in fungal pathogenesis. Understanding the specific contributions of AfpA to *A. fumigatus* physiology may reveal new aspects of transcriptional regulation in this pathogen and potentially identify novel targets for antifungal intervention.

## 2. Materials and Methods

### 2.1. Strains and Culture Conditions

All *A. fumigatus* strains used in this study, derived from the wild-type (WT) Af293 strain, were cultured on yeast extract glucose medium (YG), glucose minimal medium (MMG), or MMG with 0.1% yeast extract (MMY) with appropriate supplements, as described previously [11,12].

### 2.2. Construction of Mutant Strains

The deletion construct generated by double-joint PCR [13] containing the *Aspergillus nidulans* selective marker *AnipyrG* with the 5’ and 3’ flanking regions of the *A. fumigatus afpA* gene (AFUA_5g11390) was introduced into the recipient strains [14]. The selective marker was amplified from *A. nidulans* FGSC A4 genomic DNA. To complement the mutants, a double-joint PCR method was used [13] with *hygB* as a selective marker. The null mutants and complemented strains were confirmed using diagnostic PCR (using primer pair oligo1535/1536) followed by restriction enzyme digestion (Appendix A). The oligonucleotides used in this study are listed in Appendix A.

### 2.3. Phenotype Analyses

Radial growth was evaluated by inoculation of conidia in the center of solid media and measurement of colony diameters. Conidia production was quantified using two methods: point-inoculated cultures were used as per the growth area, and overlay-inoculated cultures were used on each plate. Conidia were collected using a 0.02% Tween 80 solution, filtered through Miracloth (Calbiochem, San Diego, CA, USA), and counted using a hemocytometer (Thermo Fisher Scientific, Waltham, MA, USA). To examine sensitivity to cell wall and oxidative stress, calcofluor white (CFW, 50 μg/mL), Congo red (50 μg/mL), and H_2_O_2_ (5 mM) were added to the YG media. Conidia (1 × 10^5^) of relevant strains were inoculated into these stress media and incubated at 37 °C. Colony diameters were measured. The inhibition ratio was calculated as follows: (colony diameter in YG media–colony diameter in stress media)/colony diameter in YG media. Chitin levels were determined based on the colorimetric assay of glucosamine in acidic hydrolysate with glucosamine-HCl as a standard [15]. The β-glucan content was quantified using the enzymatic yeast β-glucan Megazyme kit following the manufacturer’s protocol (Megazyme, Bray, Ireland). The sensitivity toward caspofungin on the growth of the WT and mutant strains was estimated using E-test strips (Biomérieux, Durham, NC, USA) as previously described [16].

### 2.4. Nucleic Acid Isolation and Manipulation

Total RNA isolation and RT-qPCR were performed as previously described [17,18,19]. Briefly, each sample was homogenized in 1 mL of TRIzol reagent (Invitrogen, Waltham, MA, USA), and the supernatant was mixed with an equal volume of cold isopropanol to precipitate RNA and centrifuged. The resulting RNA pellets were washed with 70% ethanol using diethyl pyrocarbonate-treated water and dissolved in RNase-free water. RT-qPCR was performed using AccuPower^®^ GreenStar™ RT-qPCR Master Mix (Bioneer, Daejeon, Republic of Korea) and a Rotor-Gene Q real-time PCR system (Qiagen, Hilden, Germany). cDNA synthesis was carried out at 50 °C for 15 min. PCR conditions included an initial denaturation at 95 °C for 5 min, followed by 95 °C and 55 °C for 30 s for 40 cycles. Specificity of amplification was confirmed using melting curve analysis. The expression ratios were normalized to the expression level of the endogenous reference gene *ef1*α [20,21,22,23,24] and calculated using the 2^−ΔΔCq^ method [25]. The expression stability was determined using the BestKeeper index in RefFinder (https://www.ciidirsinaloa.com.mx/RefFinder-master/, accessed on 21 January 2024) [26]. The PCR efficiencies of the studied genes were 90–102%. Primer sequences used for the target gene amplification are listed in Appendix A. For transcriptome analyses, RNA was isolated and submitted to eBiogen, Inc. (Seoul, Korea) for library preparation and sequencing. Construction of the RNA-seq library was performed using QuantSeq 3′ mRNA-Seq Library Prep Kit (Lexogen, Inc., Wien, Austria) according to the manufacturer’s instructions. High-throughput sequencing was performed as single-end 75 sequencing using NextSeq 500 (Illumina, Inc., San Diego, CA, USA).

### 2.5. Enzyme Assay

PKA activity was detected using the PepTag^®^ Non-Radioactive cAMP-Dependent Protein Kinase Assay kit (Promega, Madison, WI, USA) as previously described [6,27]. For catalase activity assays, conidia (1 × 10^5^) of relevant strains were inoculated into liquid YG with appropriate supplements and incubated at 37 °C, 250 rpm for 24 h. The mycelia were disrupted with glass beads in 20 mM phosphate buffer (pH 7.5) supplemented with a protease inhibitor cocktail. Protein content was quantified using Bradford reagent (Bio-Rad Laboratories, Inc., Hercules, CA, USA) and bovine serum albumin as a standard. Catalase activity was visualized using negative staining with ferricyanide after resolving on 15% non-denaturing PAGE in Tris-glycine buffer (pH 8.3) [28,29]. The relative intensities of enzyme activities were quantified using the Image J 1.52k software (NIH, Bethesda, MD, USA).

### 2.6. Virulence and Phagocytosis Assay

The virulence assay was conducted as previously described [4,5,30,31]. For the immunocompromised mouse model, outbred CrlOri: CD1 (ICR) mice (6–8 weeks old, weighing 30 g; Orient Bio Inc., Seongnam, Republic of Korea) were used, which were housed five per cage and had access to food and water ad libitum. Mice were immunosuppressed with subcutaneous injections of cortisone acetate at 10 mg/mouse for 4 days prior to infection, and cortisone acetate was injected subcutaneously, 10 mg/mouse, 2 days prior to infection. At days 0, 3, and 6 post-infection, administrations were repeated with cortisone acetate (10 mg/mouse). For conidia inoculation, mice were anesthetized with isoflurane and then intranasally infected with 1 × 10^7^ conidia of *A. fumigatus* strains (10 mice per fungal strain) suspended in 30 μL of 0.01% Tween 80 in PBS. Mice were monitored every 12 h for 5 days after the challenge. Control mice were inoculated with sterile 0.01% Tween 80 in PBS. For histology experiments, mice were sacrificed at day 5 after conidia infection. The lung sections of the mouse were stained with periodic acid–Schiff reagent (PAS) to determine fungal impact and hyphal growth. Kaplan–Meier survival curves were analyzed using the Log-Rank (Mantel–Cox) test for significance. A phagocytic assay was performed according to a modified method [32,33,34,35]. The MH-S cell lines were maintained in RPMI 1640 containing 10% fetal bovine serum (Invitrogen, Waltham, MA, USA) and 50 μM of 2-mercaptoethanol (Sigma, St. Louis, MO, USA). The MH-S cells were adhered to coverslips in 6-well plates at a concentration of 5 × 10^5^ cells/mL for 2 h and subsequently challenged with 1.5 × 10^6^ conidia for 1 h. Unbound conidia were removed by washing with PBS and then incubated for 2 h at 37 °C in an atmosphere of 5% CO_2_. Wells were then washed with PBS and observed using microscopy. The percentage of phagocytosis was assessed.

### 2.7. Statistical Analysis

All experiments were conducted in biological triplicate. Data are presented as mean ± standard deviation (SD), and statistical significance was set at *p* < 0.05. GraphPad Prism 4 (GraphPad Software, Inc., San Diego, CA, USA) was utilized for statistical analysis and generation of survival curves.

## 3. Results

### 3.1. Summary of AfpA in Aspergillus

Domain analysis of AfpA and its orthologs across six *Aspergillus* species revealed a conserved KilA-N domain spanning amino acids 86–171 in *A. fumigatus* (Figure 1A). This APSES family characteristic domain was consistently positioned in the N-terminal region across all examined species. While the core domain was highly conserved, some species showed additional structural features: *A. flavus* contained a unique coiled-coil domain, and both *A. clavatus* and *A. flavus* had C-terminal transmembrane regions absent in other species. The amino acid sequence of AfpA (427 amino acids) shows 66.45 (*A*. *flavus*)~96.49% (*A*. *fischeri*) identity with AfpA found in phylogenetically related *Aspergillus* species. Multiple sequence alignment of the KilA-A domain showed remarkable conservation of key residues critical for DNA binding and structural integrity (Figure 1B).

### 3.2. AfpA Negatively Influences Conidiation While Promoting Vegetative Growth

To investigate the function of AfpA in A. fumigatus, ΔafpA mutant and complemented strains were generated by double-joint PCR. The ΔafpA mutant showed distinct morphological changes on both MMY and YG media after 2 days at 37 °C (Figure 2A). Radial growth analysis revealed an approximately 20% reduction in colony diameter compared to WT and complemented strains (Figure 2B). Despite impaired growth, the ΔafpA mutant produced higher conidia per growth area than control strains (Figure 2C). In addition, RT-qPCR analysis showed up-regulation of key asexual development regulators: abaA increased 1.3-fold, while brlA and wetA increased 1.8~1.9-fold, respectively (Figure 2D). Microscopic observations revealed that the ΔafpA mutant generated immature conidiophores and did not produce conidia after 24 h incubation (Figure 3). These results indicate that AfpA is required for proper vegetative growth and asexual sporulation (conidiation).

### 3.3. AfpA Is Involved in the cAMP-PKA Signaling Pathway and Spore Germination

The cAMP-dependent protein kinase A (PKA) singling plays an important role in fungal spore germination, growth, and asexual development [36]. We, therefore, investigated the relationship between AfpA and the cAMP-PKA signaling pathway. PKA activity assays revealed that the Δ*afpA* mutant exhibited markedly decreased PKA activity, reaching only 55% of WT levels even with cAMP supplementation (Figure 4A). RT-qPCR analysis showed a reduced expression of key signaling components; *acyA* and *pkaC1* decreased 55% to 60% of WT levels (Figure 4B). Germination assays demonstrated that Δ*afpA* conidia showed delayed germination, with germ tube emergence not observed until 8 hrs, and the maximum germination rate reached only 80% (Figure 4C). These findings establish AfpA as a positive regulator of the cAMP-PKA signaling pathway.

### 3.4. AfpA Affects Cell Wall Stress Response

We investigated the impact of AfpA on the growth of A. fumigatus under cell wall stress conditions by exposing the strains to Congo red (CR). The ΔafpA mutant exhibited increased resistance to CR, showing a slightly reduced growth inhibition ratio compared to the WT and complemented strains. However, the resistance toward calcofluor white (CFW) did not change significantly by the loss of *afpA* (Figure 5A,B). To further elucidate the role of AfpA in CR resistance, we assessed the transcript levels of the key chitin biosynthesis genes chsB and chsE. Following exposure to CR (50 µg/mL for 6 hrs), the transcript levels of chsB and chsE were significantly down-regulated in the ΔafpA mutant relative to the WT and complemented strains. In contrast, the expression of gfaA, which is known to be up-regulated in response to cell wall stressors [31], was markedly elevated in the ΔafpA mutant (Figure 5C). These findings suggest that AfpA is involved in chitin biosynthesis and contributes to maintaining cell wall integrity. Furthermore, we checked the content of β-glucan and chitin of each strain. While the chitin content of the ΔafpA mutant was lower than that of WT and the complemented strain, the content of β-glucan showed no significant difference between WT and mutant strain (Figure 5D). However, the sensitivity against caspofungin was not changed distinctly by the loss of *afpA* (Appendix A).

### 3.5. Functions of AfpA in Oxidative Stress Response

Next, to investigate the role of AfpA in the oxidative stress response, we assessed the sensitivity of WT, Δ*afpA*, and complemented strains to hydrogen peroxide (H_2_O_2_), a reactive oxygen species commonly encountered by fungal pathogens during host infection. Strains were incubated on YG medium supplemented with 5 mM H_2_O_2_, and growth was monitored under oxidative stress conditions. The Δ*afpA* mutant displayed a pronounced sensitivity to H_2_O_2_, exhibiting approximately a two-fold higher growth inhibition ratio compared to the WT and complemented strains (Figure 6A,B), indicating a compromised oxidative stress defense. To determine whether this phenotype was associated with altered expression of oxidative stress-related genes, we analyzed the transcript levels of *catA* and *cat1*, which encode catalase enzymes involved in H_2_O_2_ detoxification. RT-qPCR revealed that both *catA* and *cat1* mRNA levels were significantly downregulated in the Δ*afpA* mutant relative to the control strains (Figure 6C). Consistent with the transcriptional data, native PAGE analysis of catalase activity showed markedly reduced enzymatic function of CatA and Cat1 in the mutant, being reduced to approximately 37% and 28% of WT levels, respectively (Figure 6D). Collectively, these findings indicate that AfpA is essential for regulating catalase expression and oxidative defense in *A. fumigatus*.

### 3.6. The Role of AfpA in Virulence

To evaluate the pathological significance of AfpA in fungal virulence, conidia from the WT, ΔafpA, and complemented strains were intranasally administered into immunosuppressed (neutropenic) mice, and disease progression was assessed by monitoring survival rates. Mice infected with the Δ*afpA* strain exhibited reduced survival in neutropenic mice compared with WT or complemented conidia, indicating a higher virulence potential associated with the loss of *afpA* (Figure 7A). There were no significant differences between WT and complemented strains in the survival rate. To investigate the host immune response contributing to this phenotype, we examined the interaction between fungal conidia and murine alveolar macrophages. After 4 hrs of co-culture, the rate of phagocytosis was determined by counting the proportion of macrophages that had internalized conidia. The Δ*afpA* mutant conidia showed a significantly reduced phagocytosis rate (13.6%) compared to WT and complemented strains (approximately 23%), suggesting impaired uptake by innate immune cells (Figure 7B). To further elucidate the underlying cause of increased mortality in Δ*afpA*-infected mice, lung tissues were collected from infected animals and subjected to periodic acid-Schiff (PAS) staining to evaluate fungal proliferation and tissue pathology. Histological analysis revealed a markedly higher fungal burden and more severe tissue destruction in the lungs of mice infected with the Δ*afpA* mutant compared to those infected with WT or complemented strains (Figure 7C). These findings collectively suggest that AfpA contributes to fungal pathogenicity by evading host immune response in lung tissues.

### 3.7. Transcriptome Analysis

To identify the possible function of AfpA, we performed mRNA-Seq analyses of ∆*afpA* and WT strains. Of about 14,000,000 mapped reads, 536 genes were differentially expressed more than 2-fold (*p* < 0.05), of which 295 genes were upregulated and 241 genes were downregulated (Figure 8A). In molecular function gene ontology (GO) categories, “transporter activity” and “catalytic activity” were upregulated, whereas “signal transducer activity” was downregulated (Figure 8B). As shown in Appendix A, cell wall-related genes (endo-1,3-β-glucanase, UDP-N-acetlyglucosamine 1-carboxyvinyltransferase) were up-regulated. Mitochondrial enoyl reductases, RNA-specific ribonuclease, and C6 transcription factors encoding genes were down-regulated (Appendix A).

## 4. Discussion

Transcription factors (TFs) are categorized based on their DNA-binding domains (DBDs), which interact with DNA through distinct mechanisms [37,38]. Among them, the basic helix-loop-helix (bHLH) TFs represent a major family in fungi and are characterized by their requirement to form homotypic dimers. These TFs typically bind palindromic half-sites on DNA through basic residues located at their N-termini [39,40,41]. Within this family, the APSES subfamily is the most well-characterized group in pathogenic fungi. Unique to fungi, the APSES subfamily comprises four conserved groups: orthologues of StuA, Mbp1/Swi4 and Swi6, Afp1, and Xbp1, with each group widely preserved across the fungal kingdom [7,42]. APSES TFs are integral to regulating fungal morphogenesis, metabolism, and pathogenicity. However, the function of Afp1 orthologues in many fungal species remains poorly understood.

In this study, we demonstrated that AfpA in *A*. *fumigatus* acts as a multifaceted regulator that coordinates growth, development, and virulence. The Δ*afpA* mutant displayed reduced vegetative growth but increased conidiation, producing more conidia per unit area compared to the WT and complemented strains. These findings suggest that AfpA plays a dual role in controlling both vegetative growth and asexual development. Notably, another APSES TF, StuA, is known to activate conidiation genes [43], indicating that different APSES members may have opposing roles in developmental processes. The growth defects observed in the Δ*afpA* strain are consistent with those seen in other APSES mutants, such as Δ*rgdA* and Δ*mbsA*, although the degree of phenotypic severity varies among family members [4,5].

We also discovered that AfpA is necessary for the proper cAMP-PKA signaling pathway, a central hub for controlling various cellular processes in fungi. This link provides a plausible mechanistic explanation for the wide range of phenotypes observed in the Δ*afpA* mutant, including its delayed germination. This finding aligns with previous studies on RgsA in *A. fumigatus*, which also influences this pathway, suggesting a potential crosstalk or shared regulatory network among different TFs [6].

Furthermore, the heightened sensitivity to oxidative stress in the Δ*afpA* mutant, coupled with reduced catalase expression, firmly establishes AfpA’s role in the stress response. Conversely, the increased resistance to the cell wall stressor Congo red, along with the down-regulation of chitin synthase genes, points to AfpA’s involvement in maintaining cell wall integrity. These results echo findings from other fungi, such as SsStuA in *Sclerotinia sclerotiorum*, and collectively strengthen the evidence that APSES TFs are crucial modulators of fungal stress adaptation [44].

Remarkably, the Δ*afpA* mutant exhibited increased virulence in neutropenic mice, characterized by faster mortality and more extensive tissue invasion. This hypervirulent phenotype may be linked to reduced phagocytic uptake of Δ*afpA* conidia by macrophages. In assays using murine alveolar macrophages, we observed a significantly lower percentage of Δ*afpA* conidia internalized by macrophages compared to the WT. These findings suggest that AfpA influences fungal recognition and uptake by host immune cells.

Host pattern recognition receptors (PRRs) detect specific components of the fungal cell wall—such as chitin, β-glucans, and mannoproteins—to initiate immune responses [45,46]. Interestingly, the Δ*afpA* strain showed increased resistance to Congo red and decreased expression of chitin biosynthetic genes, indicating that AfpA modulates the structural properties of the fungal cell wall. Such alterations can compromise cell wall integrity and potentially affect host–pathogen interactions [47].

Chitin, in particular, is a potent immunogenic molecule and a conserved microbe-associated molecular pattern (MAMP) relevant to fungal infections [48,49]. MAMPs like chitin activate host immunity by engaging PRRs, including Toll-like receptors (TLRs) and C-type lectin receptors (CLRs) [50,51,52]. For instance, β-glucans are recognized by the CLR Dectin-1 [52], while TLRs bind a broader range of MAMPs through their extracellular domains, signaling through the MyD88 adaptor protein. Notably, phagocytosis is specifically triggered by CLRs.

Supporting these observations, comparative transcriptomic analyses between WT and Δ*afpA* strains revealed that deletion of AfpA leads to widespread transcriptional reprogramming, affecting key biological processes such as nutrient transport, catalytic activity, and signal transduction pathways. Notably, several cell wall biosynthesis and remodeling genes were significantly up-regulated in the mutant, including endo-1,3-β-glucanase and UDP-N-acetylglucosamine 1-carboxyvinyltransferase involved in β-glucan and chitin metabolism, respectively. These expression changes align with the observed phenotypic alterations in cell wall structure and stress response, further supporting the hypothesis that AfpA modulates cell wall architecture and, by extension, fungal interaction with the host immune system.

While this study provides a comprehensive overview of AfpA’s functions, several questions remain. The exact molecular mechanisms by which AfpA regulates the cAMP-PKA pathway and its downstream targets are not yet fully understood. Future research should focus on identifying direct binding sites of AfpA to confirm its role as a master regulator. In addition, the specific changes in cell wall composition that lead to reduced phagocytic uptake also require further exploration. A detailed structural analysis of the Δ*afpA* cell wall and its components, particularly chitin and β-glucans, would provide definitive evidence of how AfpA-mediated changes affect host immune recognition.

In summary, our study identifies AfpA as a novel APSES transcription factor that integrates signals regulating growth, development, stress responses, and host–pathogen interactions. The discovery that AfpA deletion leads to increased virulence despite growth defects suggests an unexpected trade-off between development and immune evasion, mediated by changes in cell wall composition. These findings advance the understanding of APSES TF function in filamentous fungi and open new avenues for exploring fungal adaptation and immune evasion mechanisms.

## 5. Conclusions

Our studies have revealed that AfpA governs diverse biological processes with different mode from other APSES TFs, such as vegetative growth, asexual development, stress response, and virulence in the *A*. *fumigatus* for the first time. Future studies should identify AfpA’s direct transcriptional targets through ChIP-seq and investigate its interactions with other regulatory proteins to fully understand its role in coordinating *A. fumigatus* biology.

## Figures and Tables

**Figure 1 jof-11-00678-f001:**
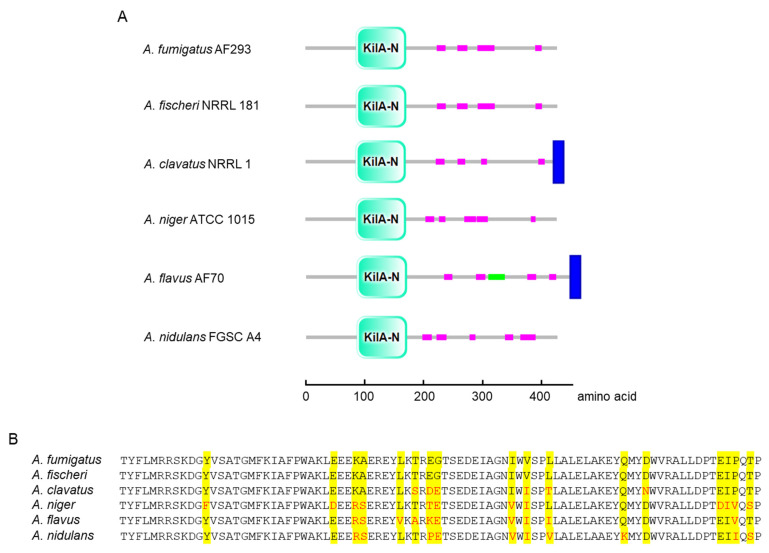
Domain architecture and amino acid sequence alignment of AfpA orthologs in *Aspergillus*. (**A**) Schematic representation of domain organization in AfpA orthologs from six *Aspergillus* species. The conserved KilA-N domain (cyan) is present in all species at similar positions (amino acids 86–171 in *A. fumigatus* AfpA). Low-complexity regions are shown in magenta, the coiled-coil domain in green (*A. flavus* only), and transmembrane regions in blue (*A. clavatus* and *A. flavus*). (**B**) Multiple sequence alignment of the KilA-N domain. Identical residues are highlighted in yellow, and functionally conserved substitutions are shown in red. Domain structures are presented using SMART (http://smart.embl-heidelberg.de (accessed on 6 May 2025)).

**Figure 2 jof-11-00678-f002:**
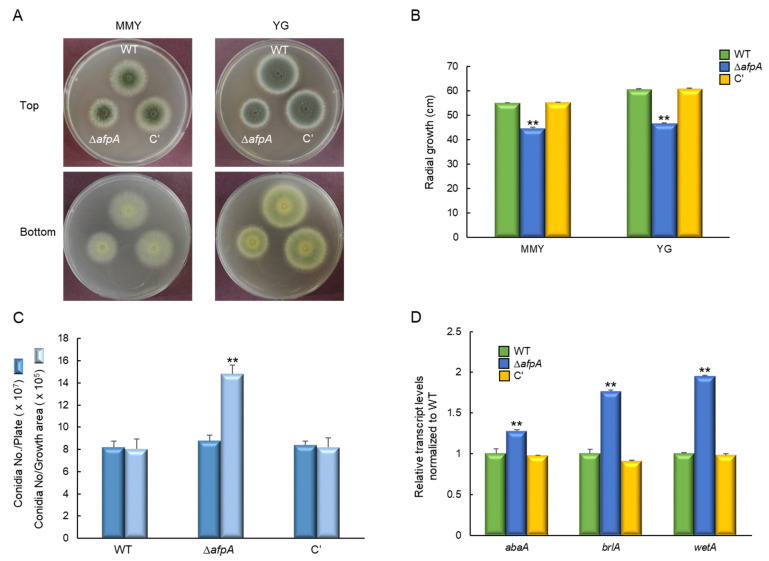
The roles of AfpA in vegetative growth and conidiation. (**A**) Colony morphology of wild-type (WT), Δ*afpA* mutant, and complemented (C′) strains grown on MMY and YG solid media for 2 days at 37 °C. (**B**) Radial growth is significantly reduced in the Δ*afpA* strain compared to WT and C′ strains on both media. (**C**) Conidia numbers produced by each strain per growth area and per plate. (**D**) Transcript levels of key asexual development regulatory genes were determined by quantitative RT-PCR (RT-qPCR). The transcript levels were normalized to the expression level of the endogenous reference gene *ef1α*. Fungal cultures were grown in MMY for 3 days, and mRNA levels were normalized to the expression level of the *ef1α* gene. Statistical significance of differences was assessed by ANOVA: ** *p* < 0.01.

**Figure 3 jof-11-00678-f003:**
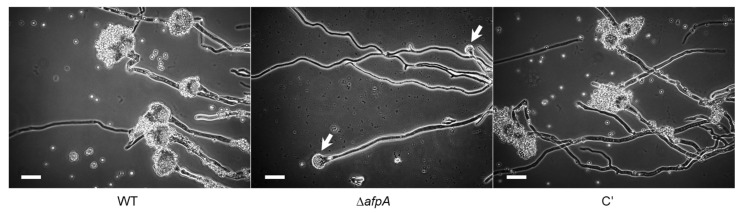
Microscopic examination of WT, mutants, and complemented (C’) strains. Conidiophores were observed under a light microscope after 24 h of incubation. Bars = 100 μm. Arrows indicate immature conidiophores.

**Figure 4 jof-11-00678-f004:**
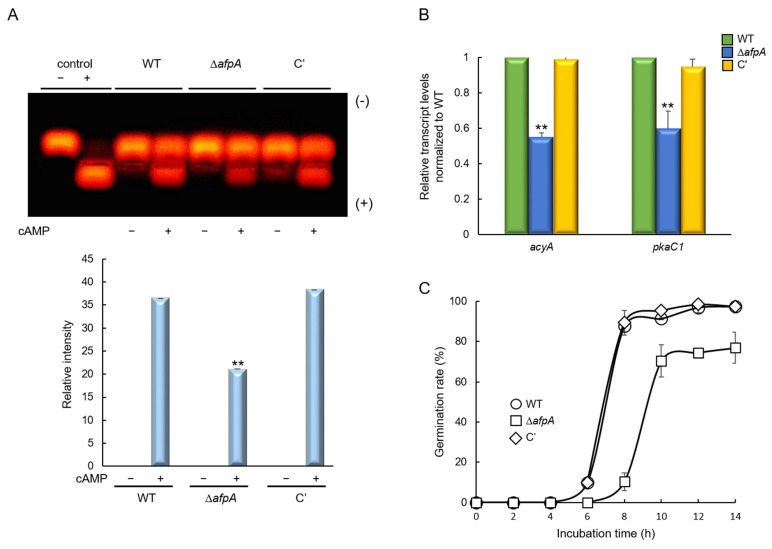
AfpA affects the cAMP-PKA signaling and timely conidia germination. (**A**) cAMP-PKA activity levels of three strains as monitored by gel electrophoresis. Control indicates the presence (+) or absence (−) of active PKA. Top; Electrophotogram of PKA activity. Bottom; Relative intensities of the enzyme activities (positive bands). (**B**) Expression levels of *acyA* and *pkaC1* mRNA in WT, Δ*afpA*, and complemented (C′) strains were analyzed by RT-qPCR. The transcript levels were normalized to the expression level of the endogenous reference gene *ef1α*. (**C**) Germination kinetics of conidia. Conidia were inoculated in MMY and incubated at 37 °C for 14 h. Statistical differences between strains were evaluated by ANOVA: ** *p* < 0.01. All experiments were performed in triplicate.

**Figure 5 jof-11-00678-f005:**
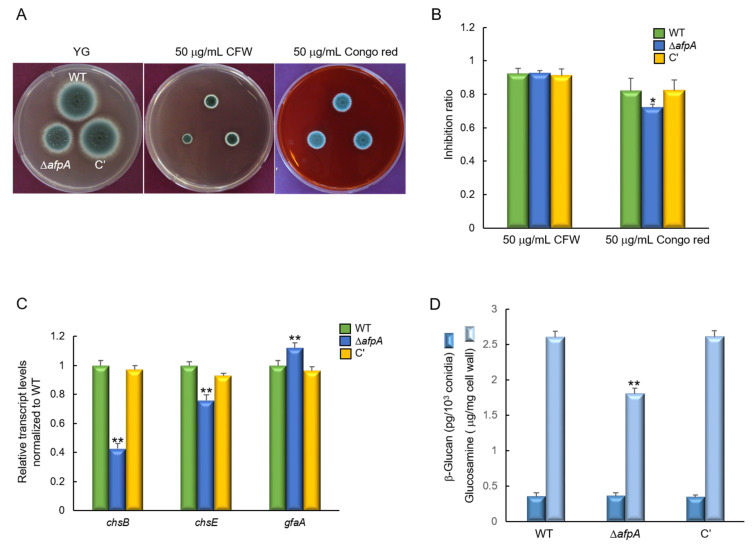
The deletion of *afpA* confers increased resistance to cell wall stress. (**A**) Colony morphology and radial growth of WT, Δ*afpA*, and C′ strains grown on YG medium with or without 50 μg/mL of calcofluor white (CFW) and Congo red for 2 days at 37 °C. (**B**) The Δ*afpA* mutant exhibits a significantly lower inhibition ratio compared to WT and C′ strains (measured in triplicate). (**C**) RT-qPCR analysis of cell wall biosynthesis genes. The transcript levels were normalized to the expression level of the endogenous reference gene *ef1α*. Expression of chitin synthase genes *chsB* and *chsE* is significantly reduced in Δ*afpA*, while expression of *gfaA* is significantly elevated. (**D**) Amount of β-glucan and chitin conidia of WT and mutant strains (measured in triplicate). Statistical significance of differences between WT and mutant strains was evaluated using ANOVA: * *p* < 0.05, ** *p* < 0.01.

**Figure 6 jof-11-00678-f006:**
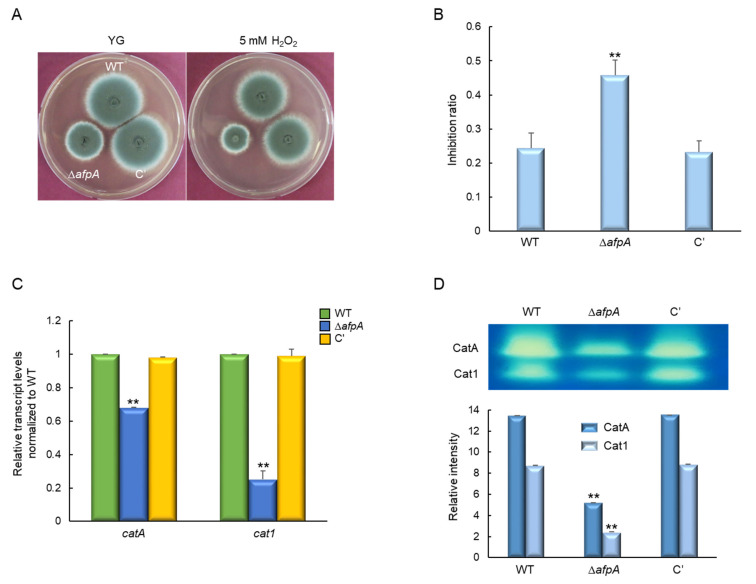
The roles of AfpA in response to H_2_O_2_ stress. (**A**) Colony appearance and radial growth inhibition of WT, Δ*afpA*, and C′ strains grown on YG agar with or without 5 mM H_2_O_2_ for 2 days at 37 °C. (**B**) The Δ*afpA* mutant shows approximately two-fold higher sensitivity to H_2_O_2_ compared to WT and C′ strains. (**C**) Both *catA* and *cat1* transcript levels are significantly reduced in the Δ*afpA* mutant strain. The transcript levels were normalized to the expression level of the endogenous reference gene *ef1α*. (**D**) CatA and Cat1 activities are markedly reduced by the loss of *afpA*. Statistical significance of differences was evaluated using ANOVA: ** *p* < 0.01. All experiments were performed in triplicates.

**Figure 7 jof-11-00678-f007:**
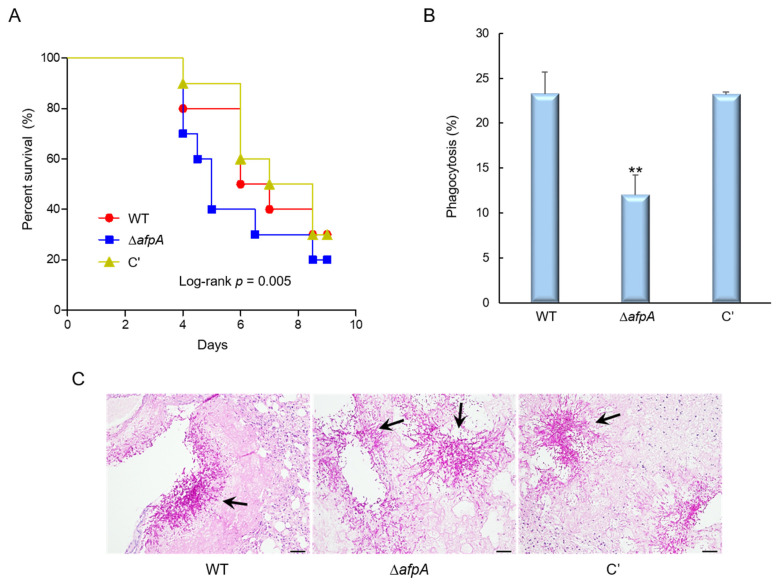
The deletion of *afpA* enhances virulence through reduced immune recognition. (**A**) Survival curves of neutropenic mice (*n* = 10 per group) intranasally infected with conidia of WT, Δ*afpA*, or C′ strains. Kaplan–Meier survival curves were analyzed using the Log-Rank (Mantel–Cox) test for significance. *p* = 0.005. (**B**) Phagocytosis indicates the percentage of macrophages containing one or more ingested conidia (*n* = 21). (**C**) Histopathological analysis of infected lungs at 3 days post-infection stained with periodic acid–Schiff reagent (PAS). Arrows indicate fungal mycelium. Scale bar = 50 μm. Statistical significance of differences between WT and mutant strains was evaluated by ANOVA: ** *p* < 0.01.

**Figure 8 jof-11-00678-f008:**
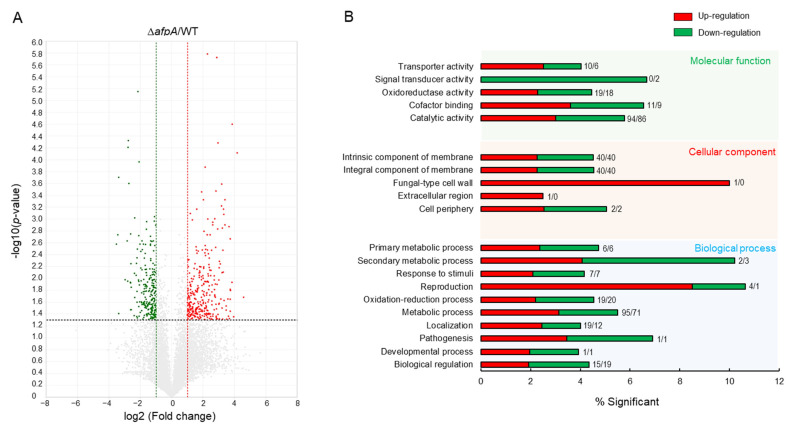
Transcriptome analyses of the Δ*afpA* strain. (**A**) Volcano plot showing the fold change (*x*-axis) and *p* value (*y*-axis) of genes sequenced in the Δ*afpA* strain compared to WT. Red and green dots denote up- and down-regulated genes, respectively. (**B**) Functional annotation histograms of DEGs in the Δ*afpA* strain. The red bars represent genes whose mRNA levels increased in the mutant, whereas the green bars represent those genes whose mRNA levels decreased in the mutant strain. Numbers represent significantly regulated gene numbers.

## Data Availability

RNA-Seq data are available from the NCBI Gene Expression Omnibus (GEO) database (GSE182799). The original contributions presented in the study are included in the article/Appendix A; further inquiries can be directed to the corresponding authors.

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
