# Peer review of "Properties of Putative APSES Transcription Factor AfpA in Aspergillus fumigatus"

_jof, 2025, doi:10.3390/jof11090678_

Round 1

Reviewer 1 Report

The manuscript by Choi et al. presents a comprehensive functional characterization of AfpA, an APSES-type transcription factor in Aspergillus fumigatus. The authors generated a ∆afpA mutant and demonstrated its role in regulating vegetative growth, asexual development, stress responses, and virulence. The experimental design is sound, and the data are generally well presented. However, several aspects of the manuscript require clarification or revision to enhance clarity and scientific rigor. Detailed comments are as follows:

  1. It would be more informative to present the mutant construction strategy in a schematic figure (either in main or supplementary figure), along with experimental confirmation of ∆afpAand the complemented strain (C’) by PCR. Validation of these strains using Southern blot or RT-qPCR would further strengthen the conclusions.
  2. Lines 97–112: There is no indication that genomic DNA was removed from RNA preparations. Include the DNase treatment step in the RNA isolation protocol to show the accuracy of downstream RT-qPCR analysis.
  3. Line 129: Provide the concentrations and dosing regimens of cyclophosphamide and cortisone acetate used for immunosuppression in the mice model.
  4. The Materials and Methods section lacks a description of the native PAGE method used for analyzing catalase activity.
  5. In RT-qPCR data, stability of the reference geneef1α under the tested stress and developmental conditions is debatable. Provide a validated reference that supports the use of ef1α as a housekeeping gene in  fumigatus. References 18 and 19 do not support its use in A. fumigatus.
  6. The finding that the ∆afpAstrain exhibits increased virulence (Fig. 6), potentially via reduced immune recognition, is intriguing. However, deeper mechanistic insights are needed. An additional data would be valuable such as flow cytometry or confocal microscopy to quantify surface β-glucan and chitin exposure in WT, ∆afpA, and C’ strains.
  7. The enhanced Congo red (CR) resistance observed in ∆afpA, along with increased virulence, raises the question of whether cell wall alterations contribute to immune evasion by masking PAMPs. Discuss this possibility in the context of relevant literature with references.
  8. All figures should clearly indicate the number of biological repeats and technical replicates used in each experiment.
  9. Ensure consistent terminology for strain designations throughout the manuscript. At times, the ∆afpAmutant is referred to as the “afpA null mutant.”

The manuscript by Choi et al. presents a comprehensive functional characterization of AfpA, an APSES-type transcription factor in Aspergillus fumigatus. The authors generated a ∆afpA mutant and demonstrated its role in regulating vegetative growth, asexual development, stress responses, and virulence. The experimental design is sound, and the data are generally well presented. However, several aspects of the manuscript require clarification or revision to enhance clarity and scientific rigor. Detailed comments are as follows:

  1. It would be more informative to present the mutant construction strategy in a schematic figure (either in main or supplementary figure), along with experimental confirmation of ∆afpAand the complemented strain (C’) by PCR. Validation of these strains using Southern blot or RT-qPCR would further strengthen the conclusions.
  2. Lines 97–112: There is no indication that genomic DNA was removed from RNA preparations. Include the DNase treatment step in the RNA isolation protocol to show the accuracy of downstream RT-qPCR analysis.
  3. Line 129: Provide the concentrations and dosing regimens of cyclophosphamide and cortisone acetate used for immunosuppression in the mice model.
  4. The Materials and Methods section lacks a description of the native PAGE method used for analyzing catalase activity.
  5. In RT-qPCR data, stability of the reference geneef1α under the tested stress and developmental conditions is debatable. Provide a validated reference that supports the use of ef1α as a housekeeping gene in  fumigatus. References 18 and 19 do not support its use in A. fumigatus.
  6. The finding that the ∆afpAstrain exhibits increased virulence (Fig. 6), potentially via reduced immune recognition, is intriguing. However, deeper mechanistic insights are needed. An additional data would be valuable such as flow cytometry or confocal microscopy to quantify surface β-glucan and chitin exposure in WT, ∆afpA, and C’ strains.
  7. The enhanced Congo red (CR) resistance observed in ∆afpA, along with increased virulence, raises the question of whether cell wall alterations contribute to immune evasion by masking PAMPs. Discuss this possibility in the context of relevant literature with references.
  8. All figures should clearly indicate the number of biological repeats and technical replicates used in each experiment.
  9. Ensure consistent terminology for strain designations throughout the manuscript. At times, the ∆afpAmutant is referred to as the “afpA null mutant.”

Reviewer 2 Report

Overall, the manuscript is well-written and provides valuable insights into the role of AfpA. The authors conducted a series of experiments that contribute to the understanding of AfpA function. The Materials and Methods section is clearly described, and the Results section is generally easy to follow. However, several figures require further clarification, and some minor revisions are necessary. The Discussion is concise and appropriately interprets the findings.

The manuscript is suitable for publication after minor revisions.

  1. Supplementary Materials:
  • Please provide descriptive titles and captions for the supplementary figures.
  • Table S1 is referenced but not included in the supplementary materials. Please, ensure it is provided for review.
  • The resolution of the two supplementary figures is low. Please provide higher-quality images. Also, consider including relative intensity values where applicable to enhance data interpretation.

2. Typo Errors:

Line 172: "AfpA" is incorrectly written as "AtfA". Please, correct.

Line 173: There is a red dot in this line. Please, make it black.

Line 180: The word “required” is misspelled as "requred". Please correct it.

3. Figure 3:

The legend for figures 3 should be revised. Panel A includes both top and bottom parts, but this distinction is not clearly mentioned in the legend. Please revise to provide a clear explanation for each panel.

4. Statistical Analysis:

The manuscript consistently uses Student’s t-test, including in cases where more than two groups are compared (e.g., WT, mutant, and complement strains). In such cases, ANOVA followed by appropriate post hoc tests would be more statistically appropriate. Please revise the analysis or justify the use of Student’s t-test in these contexts.

4. Additional Suggestion:

Since AfpA is proposed to be involved in cell wall function, did the authors consider testing the sensitivity of the mutant strains to antifungal agents that target cell wall synthesis (e.g., caspofungin)? If such data are available or planned, it would strengthen the functional relevance of the findings.

Reviewer 3 Report

Young-Ho Choi et al. present an article on the initial characterization of a putative transcription factor in A. fumigatus, AfpA, investigating its role in vegetative growth, spore germination, cAMP-PKA signaling, oxidative and cell wall stress responses and finally in A. fumigatus virulence. Results are discussed in the context of comparing with other similar transcription factors.

In general, the article contains some interesting information, in my opinion however, the offered results are rather preliminary in order to justify publication in a high impact journal like JoF. In a previous publication, the authors have shown that another APSES-like transcription factor, MbsA, affects spore germination rates and lowers expression of genes associated with key conidiation genes abaA, brlA, and wetA, chitin synthesis and the SakA/PKA pathway. Moreover, ΔmbsA strains showed reduced virulence, likely due to the defective spore integrity. The experimental approach and the presentation/interpretation of results for MbsA are similar to ones presented herein for AfpA. More in depth analysis on the actual role of this transcription factor is therefore required, as also mentioned in the conclusion.

Lines 38-63: These paragraphs need some rewriting/restructuring to better explain why AfpA belongs to the APSES members.

Chapter 3.2: The ΔafpA mutant phenotypes should also be checked and compared at the microscopic level, along with the appropriate controls.

Lines 285-302: This part is a repetition of the information offered in the introduction and should thus be shortened.

Lines 303-335: This part is a repetition of the results. The entire discussion part should actually be rewritten emphasizing more on the importance and novelty of this work

Round 2

Reviewer 1 Report

I recommend acceptance of the manuscript in its current form, as no further revisions are necessary.

I recommend acceptance of the manuscript in its current form, as no further revisions are necessary.

Author Response

I recommend acceptance of the manuscript in its current form, as no further revisions are necessary.

⇒ Thank you very much for your recommendation.

Reviewer 3 Report

I thank the authors for responding to some of the comments of the previous assessment. The article has been improved. However, in my opinion, the data presented are still preliminary, requiring a more in depth analysis on the actual role of this transcription factor.

n.a.

Author Response

I thank the authors for responding to some of the comments of the previous assessment. The article has been improved. However, in my opinion, the data presented are still preliminary, requiring a more in depth analysis on the actual role of this transcription factor.

             ⇒ Thank you for your valuable comment. We added new transcriptomic data in the text and Figure 8, Table S2, and Table S3. (Line 120-125, 321-336, 423-434)

Round 3

Reviewer 3 Report

I have no further comments.

na

Author Response

Comment: I have no further comments.

Response: Thank you for your comment.